# Factors Influencing the Maturation and Developmental Competence of Yak (*Bos grunniens*) Oocytes In Vitro

**DOI:** 10.3390/genes14101882

**Published:** 2023-09-27

**Authors:** Luoyu Mo, Jun Ma, Yan Xiong, Xianrong Xiong, Daoliang Lan, Jian Li, Shi Yin

**Affiliations:** 1College of Animal and Veterinary Sciences, Southwest Minzu University, Chengdu 610041, China; moluoyu20@163.com (L.M.); jinguicilang2306@163.com (J.M.); xiongyan0910@126.com (Y.X.); xianrongxiong@163.com (X.X.); landaoliang@163.com (D.L.); lijian@swun.cn (J.L.); 2Key Laboratory of Qinghai-Tibetan Plateau Animal Genetic Resource Reservation and Utilization, Ministry of Education, Qinghai-Tibetan Plateau Animal Genetic Resource Reservation and Utilization Key Laboratory of Sichuan Province, Chengdu 610041, China; 3Key Laboratory of Animal Science of National Ethnic Affairs Commission of China, Southwest Minzu University, Chengdu 610041, China

**Keywords:** yak, oocyte, in vitro maturation, embryo development

## Abstract

The yak (*Bos grunniens*) is a unique breed living on the Qinghai–Tibet Plateau and its surrounding areas, providing locals with a variety of vital means of living and production. However, the yak has poor sexual maturity and low fertility. High-quality mature oocytes are the basis of animal breeding technology. Recently, in vitro culturing of oocytes and embryo engineering technology have been applied to yak breeding. However, compared to those observed in vivo, the maturation rate and developmental capacity of in vitro oocytes are still low, which severely limits the application of in vitro fertilization and embryo production in yaks. This review summarizes the endogenous and exogenous factors affecting the in vitro maturation (IVM) and developmental ability of yak oocytes reported in recent years and provides a theoretical basis for obtaining high-quality oocytes for in vitro fertilization and embryo production in yaks.

## 1. Introduction

The yak is a unique breed that lives in the Qinghai–Tibet Plateau and offers various necessities for local people’s survival and production. However, the reproductive performance of the yak is low. Yaks are generally calved every other year, with one in two years or two in three years, and the average annual reproductive rate for female yaks is less than 60% [1,2]. The yak has low reproductive performance due to various factors, including seasonal breeding, delayed puberty, and a low frequency of estrus [2,3,4,5].

In order to improve the reproductive performance of yaks, assisted reproductive procedures such as in vitro fertilization (IVF), somatic cell nuclear transfer (SCNT), and embryo transfer (ET) have been widely used in yak breeding [6,7,8,9]. These technologies cannot be implemented without high-quality mature oocytes. The process of oogenesis can be divided into three main stages: The first stage is the proliferation stage. This stage begins with the migration of primordial germ cells to the embryonic reproductive ridge that has not yet begun to differentiate. At this time, the female germ cells are called oogonia, and the oogonia carry out mitosis and proliferate continuously. In the middle and late stages of embryonic development, some oogonia begin to accumulate nutrients, grow, and undergo the first meiosis (MI). These cells are called primary oocytes. Primary oocytes go through the leptotene, zygotene, and pachytene stages, and finally arrest in the diplotene stage. The second stage is the growth stage, in which the volume of oocytes increases significantly, the number and shape of organelles changes greatly, and many nutrients, such as protein, carbohydrates, and lipid droplets are accumulated in the cytoplasm. The third stage is the maturity stage [10]. The oocyte completes its growth and approaches the size of the mature oocyte in volume. At this time, the arrested oocyte has an intact nuclear envelope known as the germinal vesicle (GV). Upon sexual maturity, primary oocytes resume meiosis by overcoming the effect of oocyte maturation inhibitors secreted by granulosa cells under the influence of luteinizing hormone (LH) peaks. After that, a series of changes, such as germinal vesicle break down (GVBD), chromosome aggregation, and uniform distribution of organelles in the cytoplasm occurs, and the primary oocyte subsequently completes the first meiosis, produces a secondary oocyte with half the number of chromosomes, and extrudes a first polar body. Then, the oocyte arrests again in the second meiotic metaphase (MII) and waits for fertilization. Sperm penetration causes the second polar body to extrude and the formation of a diploid fertilized egg, initiating the development of the fertilized egg, cleavage, and blastocyst formation [11,12,13].

A series of drastic morphological and molecular changes occur during oocyte maturation, such as the change in transcription, the repositioning of organelles, and the storage and utilization of maternal materials. Therefore, oocyte maturation is a complex process influenced by several internal and external influences [14,15,16]. Despite researchers continuously trying to improve the culture system of mature oocytes, the maturity rate of oocytes in vitro remains lower than that in vivo [17,18]. The low effectiveness of oocyte maturation in vitro and poor embryo quality seriously restrict the utilization of in vitro fertilization and embryo production in the yak. This review summarizes numerous recently reported internal and external factors affecting yak oocytes’ maturation and developmental competence in vitro, providing a theoretical basis for obtaining high-quality oocytes for fertilization and embryo production.

## 2. Samples

### 2.1. The Development Stage of Oocytes

Oocyte quality and maturation rate vary depending on follicle type and morphology. Liu Ben et al. found no noticeable difference in the cleavage rates of embryos after fertilization between oocytes from pre-estrus calves and adult yaks. However, the blastocyst rate of fertilized embryos from pre-estrus calves was lower than that those from adult yaks [19]. Yan Ping et al. discovered that the in vitro maturation rate of follicular phase oocytes was significantly higher than that of luteal phase oocytes [20]. The maturation and cleavage rates of ovarian surface oocytes were significantly higher than those of ovarian oocytes [21]. Li Tianqiang et al. compared different types of follicles and discovered that the number of available cumulus–oocyte complexes (COCs) and the maturity rate of oocytes obtained from medium follicles were higher than those of small, large, and engorged follicles [22].

### 2.2. The Layers of COCs

According to the criteria established by Wang Yuheng et al. for the yak oocyte classification, yak COCs were divided into four grades [23]. The results of the four types of COC culture revealed that the oocyte maturation rate was higher in grade A and B COCs [23]. Sun Yonggang et al. also graded COCs and cultured them according to the above standard. The results showed that grade A COCs had the highest maturity, cleavage, and blastocyst rates. This indicated that the number of layers of cumulus cells had an important influence on oocyte maturation and the degree of embryo development in vitro [6].

## 3. Endogenous Factor

### 3.1. Dynamic Transcripts and Protein Changes

The maturation and development of oocytes are complicated processes involving the regulation of various genes and proteins [15,24]. Dynamic changes of transcripts and proteins were observed during the development and maturation of yak oocytes [25,26]. Pei Jie et al. constructed the molecular structure of yak ovarian cortex cells using single-cell RNA sequencing (scRNA-seq). They identified the molecular features and biological functions of different cell populations. Differentially expressed genes (DEG) between oocytes and other types of ovarian cells were mainly enriched in cell cycle transition, DNA repair, and chromosome segregation processes, according to Gene Ontology (GO) enrichment analysis. Gene expression specificity testing indicated that the characterized genes *CENPF*, *TOP2A*, *MIS18BP1*, *FST*, and *INHA* were highly expressed in yak oocytes. The *FST* and *TOP2A* genes could be considered as the molecular features of yak oocytes within primordial follicles. They also discovered that the yak oocytes regulated the other types of ovarian cells primarily via the interaction between the ligands FAM3, INHA, and JAG1 with their corresponding receptors. However, the endothelial, epithelial, and granulosa cells regulated the oocytes principally via the BMP family [27]. 

Lan Daoliang et al. sequenced the transcriptome of MII stage yak oocytes and analyzed the primary biological functions and signaling pathways involved in the differentially expressed genes. KEGG analysis revealed that actin cytoskeleton regulation was the most significantly enriched pathway. the cell cycle, tight junction, adhesion patch, MAPK signaling, Wnt signaling, and endocytosis pathways were effectively enriched. The results suggested that yak oocyte maturation involved a series of dynamic processes that was highly spatially and temporally regulated [28]. Moreover, they comparatively analyzed the transcriptome data of yak oocytes matured in vitro at the MII and GV stages. They screened 4767 DEGs, of which 1418 were up-regulated and 3349 were down-regulated. Apoptotic, endocytosis, and metabolic pathways were the primary key regulatory pathways related to the up-regulated genes, while metabolic, pyruvate metabolism, focal adhesion, and oxidative phosphorylation were the key regulatory pathways related to down-regulated genes. Additionally, they discovered that immune-related interleukin family and interferon family genes were significantly differently expressed during the IVM of yak oocytes [29].

Zhao Yanling et al. applied the liquid chromatography tandem mass spectrometry (LC-MS-MS) technique to construct the protein expression profiles of yak oocytes at the GV stage. They identified 550 proteins, of which MVP, PDIA3, HSP90B1, HSPA5, and BSA were the proteins with the highest abundance expression in yak oocytes. GO analysis revealed that the proteins related to cell components were mainly concentrated in the cytoplasm (125 proteins). In the enrichment analysis related to molecular function, the number of proteins that perform the function of poly A binding RNA was the highest (54 proteins), In enrichment analyses related to biological processes, the number of proteins involved in the negative regulatory process of endopeptidase activity was the highest (26 proteins). According to KEGG analysis, the representative pathways include the complement system, ribosome metabolism, phosphatidylinositol 3 kinase-serine/threonine kinase (PI3K-Akt) signaling, and carbon metabolism pathways. Additionally, protein network analysis revealed that ALB, GAPDH, HSP90AB1, ACLY, ACTA1, HSPA8, ACTG1, UBA52, PLG, and EEF2 were the key nodes of protein–protein interactions (PPI) in oocytes [30]. Huang Yong applied isobaric tags for relative and absolute quantitation (iTRAQ) and high-performance liquid chromatography–electrospray tandem mass spectrometry (LC-ESI-MS-MS) techniques to characterize the proteins of yak GV and MII oocytes, identifying 1188 proteins. GO and KEGG analyses showed that these proteins were involved in 263 signaling pathways. Among these pathways, the metabolic pathway was the most significantly enriched. Furthermore, the research discovered that these proteins were mainly involved in protein post-translational modification, conversion, molecular chaperoning, and other general functions. Moreover, this research identified 161 differentially expressed proteins between two stages, with 81 being up-regulated and 80 being down-regulated. These differentially expressed proteins were also most significantly enriched in the metabolic pathway, followed by the actin cytoskeleton pathway [31].

### 3.2. Epigenetic Regulations

Epigenetic regulation is a kind of regulation of gene expression by changing non-gene sequence, mainly including DNA methylation, histone modification (acetylation, methylation, phosphorylation, etc.), and regulation of non-coding RNA, etc., which is a hot research topic in different fields of biology [32,33]. The recent progress in the study of epigenetic regulations on the maturation and developmental competence of yak oocytes in vitro is as follows:

#### 3.2.1. DNA Methylation

During mammalian development, DNA methylation plays a crucial role in genomic imprinting, transposon silencing, X-chromosome inactivation, and tissue-specific gene expression regulation [34]. Li Qin found that Aflatoxin B1 (AFB1) exposure caused early apoptosis in yak oocytes and affected the DNA methylation levels of mature oocytes. These effects were specifically demonstrated by the fact that AFB1 exposure significantly enhanced the fluorescence signal of DNA methylation, enhanced the mRNA transcript abundance of the methylation transferase DNMT1, and decreased the mRNA transcript abundance of *DNMT3a* and *DNMT3b*. It is well known that the major function of the methyltransferases DNMT3a and DNMT3b is to establish DNA methylation, and the methyltransferases DNMT1 play a major role in the subsequent maintenance of genome-wide methylation patterns [35,36,37]. These results suggested that AFB1 exposure was detrimental to the establishment and maintenance of DNA methylation during yak oocyte maturation. In contrast, the addition of 50 µg/mL AA during the IVM of yak oocytes attenuated these impairments, restored the mRNA transcript abundance of *DNMT1* and *DNMT3a*, and attenuated the DNA methylation [38].

In addition, adding 50 µg/mL AA increased the mRNA transcript abundance of *DNMT3a* and ten-eleven translocation enzyme (*TET3*) at the 2-cell, 4-cell, 8-cell, and morula stages of yak embryos. At the blastocyst stage, the promoter region of pluripotency gene *NANOG* exhibited moderate methylation. Addition of 50 µg/mL AA decreased the methylation level of the *NANOG* promoter, and significantly increased the mRNA transcript abundance of *NANOG*. These results suggested that 50 µg/mL AA might regulate the formation of yak blastocysts via regulating the methylation and expression of *NANOG* [38,39].

#### 3.2.2. Histone Modification

Histone modification refers to altering the interaction of histones with DNA by adding or removing specific chemical markers on histones, thereby affecting the gene transcription regulation and the occurrence of other cellular processes [40,41]. Several histone modifications, including methylation, acetylation, ubiquitination, and phosphorylation, are essential for oocyte maturation [42,43,44,45].

KDM1A is a histone demethylase with specificity for H3K4me2/me1 demethylation [46]. Han Jie et al. discovered that the *KDM1A* mRNA expression decreased and then increased during IVM of yak oocytes, with the MI stage exhibiting the lowest *KDM1A* expression. After repressing the expression of *KDM1A* in yak oocytes by adding GSK-KDM1A, the specific inhibitor of KDM1A, the cumulus cell expansion, the first polar body excretion rate, and the cleavage rate of yak oocytes were significantly decreased. The expressions of *OCT4* and *SOX2* were significantly up-regulated after the treatment. The results suggested that KDM1A was involved in regulating the meiosis of yak oocytes, possibly influencing oocyte maturation and embryo development [47,48].

KDM2 is a demethylase with a JmjC domain [49]. Cai Wenyi investigated the function of KDM2A and KDM2B in yak oocytes and discovered that *KDM2* was dynamically expressed during oocyte meiosis. Its mRNA/protein expression level was significantly higher in the MI stage than in the GV and MII stages. Moreover, adding butyrazide, a specific inhibitor of KDM2, to the in vitro maturation medium reduced oocyte maturation but increased the cleavage and blastocyst rates, suggesting that KDM2 may play opposing roles in regulating oocyte maturation and embryo development in yaks [50].

SIRT1 is a class-III nicotine adenine dinucleotide (NAD)-dependent histone deacetylases (HDACs) that have been proven to be essential for delaying oocyte aging and promoting oocyte maturation in mice [51,52]. Xiong Xianrong et al. presented that SIRT1 was involved in the IVM of yak oocytes by adding the specific agonist SRT2104 and the inhibitor inauhzin of SIRT1 to the in vitro medium of yak COCs. The results disclosed that adding SRT2104 to oocytes alleviated aging and improved the rate of IVM and the developmental ability of early embryos. In contrast to the inauhzin treatment group, SRT2104 addition significantly increased the degree of cumulus cell expansion and the cleavage and blastocyst rates of IVF embryos. It significantly inhibited the reactive oxygen species (ROS) level in oocytes by up-regulating *SIRT1*, *FOXO3a*, and *SOD2* mRNA expressions and down-regulating *BAX* mRNA expression [53].

#### 3.2.3. miRNA

miRNAs are a class of small non-coding RNAs that play a regulatory role at the post-transcriptional level by recognizing binding sites in the untranslated regions of target genes, leading to mRNA degradation or protein translation inhibition [54,55]. Recently, miRNAs have played important roles in mammalian oocyte maturation, follicular development, and early embryonic development [56,57,58].

Xiong Xianrong et al. sequenced miRNAs in yak GV and MII stage oocytes and obtained 801 and 1018 known miRNAs, respectively. Seventy-five of these miRNAs were significantly differentially expressed, with 47 miRNAs exhibiting up-regulated expression and 28 miRNAs exhibiting down-regulated expression in MII stage oocytes compared to the GV stage. Moreover, they analyzed the expression patterns of miR-16 and miR-342 in GV and MII oocytes and preimplantation embryos (two-cell, four-to-eight-cell, and blastocyst). The results revealed that miR-16 was highly expressed in GV oocytes but significantly decreased from the MII stage to the blastocyst stage. However, miR-342 expression was significantly higher in MII oocytes than in GV oocytes and significantly decreased from the two-cell embryo to the blastocyst stage. Subsequently, the miR-16 and miR-342 potential target gene expressions, such as *BCL2*, *CCND1*, *DNMT1*, and *BMP7* were analyzed in GV oocytes, MII oocytes, and preimplantation embryos. The results showed that *BCL2*, *CCND1*, and *BMP7* were expressed in the opposite direction to miR-16 and miR-342, while *DNMT1* had no defined expression pattern throughout the developmental stages [59].

Xiong Xianrong et al. studied the role of miR-342-3p in the meiotic maturation of yak oocytes. They discovered that miR-342-3p inhibitor effectively up-regulated *DNMT1* expression and significantly inhibited oocyte meiotic maturation by targeting the 3′-untranslated regions (UTR) of *DNMT1* [60]

### 3.3. G Protein-Coupled Receptor 50 (GPR50)

GPR50 is an orphan G protein-coupled receptor on the X chromosome [61]. Previous research revealed that GPR50 was strongly expressed in yak brain, ovary, and testis tissues, implying that GPR50 might have a function in reproductive development [62]. Yao Ying et al. found that the GPR50 protein was centrally expressed in the membrane during the germinal vesicle (GV) phase of yak oocytes, with the highest GPR50 expression level during the MII phase [62]. Based on these observations, researchers investigated the impacts of *GPR50* knockdown and overexpression on yak oocytes. The results indicated that the *GPR50* knockdown significantly reduced the oocyte maturation rate and the polarbody excretion rate, while *GPR50* overexpression exerted no significant influence on the excretion rate and maturity level of the yak oocytes, suggesting that GPR50 might play a crucial role in yak oocyte maturation in vitro [63].

The effects of some endogenous factors on yak oocytes and the possible mechanisms are shown in Table 1.

## 4. Exogenous Factor

### 4.1. Growth Factor

Growth factors are a class of peptides that regulate multiple effects, such as cell growth and other cellular functions, by binding to specific, high-affinity cell membrane receptors. They play critical roles in resuming oocyte meiosis, oocyte maturation, and follicular development [64,65]. Moreover, growth factors can significantly improve the development ability of embryos and promote blastocyst formation, which has great biological effects on the development processes and different developmental periods of mammalian embryos [66,67,68]. 

Several important growth factors, such as epidermal growth factor (EGF), insulin-like growth factor I (IGF-1), fibroblast growth factor 10 (FGF10), and leukemia inhibitory factor (LIF) are essential for oocyte maturation and embryo development. EGF can promote cell proliferation, differentiation, and mammalian oocyte maturation [64,69,70,71]. Ma Li et al. discovered that supplementation with 40 μg/mL EGF could significantly improve oocyte maturation and the development ability of parthenogenetic embryos [72]. Pan Yangyang, et al. found that the medium supplemented with 100 ng/mL EGF could significantly increase the yak COC maturation rate, and cleavage and blastocyst rates after fertilization. This might be caused by the inhibition of EGF on the expression of the pro-apoptosis gene *BAX* and the promotion of EGF on the expression of anti-apoptosis genes *BI-1* [73].

IGF-1 belongs to the insulin-like growth factor family [74], which is involved in mediating cellular proliferation, differentiation, and apoptosis, and plays a critical role in mammals’ growth and development [75]. Pan Yangyang et al. discovered that adding 100 ng/mL IGF-1 to the culture medium significantly increased the yak oocyte maturation rate in vitro and the cleavage and blastocyst rates of chemically activated embryos. Further study proved that this result was due to the up-regulation of IGF-1-induced cold-inducible RNA-binding protein (CIRP) [76].

FGF10 is a paracrine fibroblast growth factor involved in numerous biological processes, including embryonic development, cell growth, morphogenesis, tissue repair, tumor growth, and invasion [77,78,79,80]. It is also involved in follicle development and oocyte maturation in various mammals [81,82]. Pan Yangyang et al. found that adding 5 ng/mL of FGF10 to the culture medium of yak COCs improved the yak oocyte maturation rate and fertilization ability. This positive effect was achieved via the up-regulation of FGF10 on *CD9*, *CD81*, *DNMT1* and *DNMT3B* expressions in COCs to optimize sperm–egg interaction and DNA methylation during fertilization [6].

LIF is a potent cytokine in the IL-6 family of cytokines [83]. Zhao Tian et al. indicated that adding LIF during the IVM of yak oocytes improved oocyte quality, maturation competence, blastocyst quality, and oocyte development. The addition of LIF (50 ng/mL) to the maturation medium could increase the maturation rate and significantly lower ROS generation and the apoptosis levels of oocytes by increasing the mRNA transcription levels of anti-apoptotic and antioxidant-related genes *BCL2*, *CAPASE3*, *SURVIVIN*, *SOD2* and *GPX4* in yak oocytes. Furthermore, blastocysts formed from 50 ng/mL LIF-treated oocytes had higher total cell numbers and lower apoptosis rates than the control group [84].

### 4.2. Antioxidants

ROS are small molecules produced by biological aerobic metabolism, an include superoxide, peroxide, and oxygen radicals [85]. Reactive oxygen molecules are chemically reactive due to extra-nuclear unpaired electrons. Excessive ROS attack intracellular small molecules, such as lipids, proteins, and nucleic acids, leading to DNA degradation in the nucleus and mitochondria, causing intracellular protein denaturation, inactivating some important enzymes, inducing cellular plasma peroxidation, and ultimately triggering cell apoptosis [86]. High ROS levels accelerated the oocyte senescence, reduced oocyte quality, and caused oocyte apoptosis [87,88]. 

Vitamin A is an indispensable nutrient that regulates physiological processes such as reproduction, embryonic development, vision, growth, cell differentiation, and proliferation [89]. Vitamin A can be oxidized to retinoic acid (RA) via oxidation reactions, and RA functions as a gene expression regulator [90]. Vitamin A regulates oocyte maturation via typical and atypical signaling pathways [91,92]. According to studies, adding 2 μM Vitamin A to yak oocytes in vitro maturation medium significantly increased the rate of IVM and parthenogenetic activation (PA) embryo cleavage rate. The expressions of *STRA8*, *RARA*, and *RXRA* were highest in the MII stage compared with those in the GV and MI stages under the treatment of 2 μM Vitamin A. Additionally, the mRNA expressions of several genes in the typical signaling pathway, including *RXRA*, *RARA*, and *STRA8*, were significantly higher than those of *MEK* and *MEK1*, which were node genes of the atypical signaling pathway. These results suggested that RA was mainly dependent on the typical signaling pathway for the yak oocyte development in vitro [93,94].

Vitamin C (ascorbic acid) is a strong water-soluble antioxidant that can catalyze the reduction of oxidized glutathione to reduced glutathione [95]. Exposure of yak oocytes to 1 nM Aflatoxin B1 (AFB1) induced early oocyte apoptosis and increased intracellular ROS levels. It caused incomplete actin and uneven distribution of mitochondria, resulting in decreased quality of mature yak oocytes. However, adding 50 μg/mL Vitamin C to the culture medium protected yak oocytes from the toxic effects of AFB1 exposure. Specifically, 50 μg/mL Vitamin C reduced intra-oocyte ROS levels, repressed early oocyte apoptosis, improved mitochondrial distribution status, and restored actin distribution [38,96].

The addition of antioxidants to oocytes and embryos during in vitro culture is necessary to maintain normal levels of intracellular ROS. Melatonin is a natural endogenous indole hormone produced by the mammalian pineal gland [97]. Since melatonin is fat- and water-soluble, it can easily transfer hydrogen and electrons across cell membranes, directly scavenging free radicals and reducing cellular ROS levels [86]. Peng Wei et al. investigated the effects of melatonin on the IVM of yak oocytes by adding different concentrations of melatonin to the culture medium of yak COCs. They discovered that adding 10^−9^ M melatonin could significantly increase the oocyte maturation rate, IVF embryo cleavage and blastocysts rates, and GSH content of oocytes and blastocysts. Reductions of ROS levels, mitochondrial protein extent, DNA damage, and cell apoptosis were observed after melatonin treatment. Additionally, 10^−9^ M melatonin repaired the spindle mismatch and chromosomal abnormalities caused by oxidative stress. The results suggested that 10^−9^ M melatonin addition could alleviate oxidative stress during the IVM of oocytes and improve the oocyte maturation rate and the developmental ability of subsequent embryos [98,99].

Li Ruizhe compared the effects of antioxidants, including melatonin, Vitamin C, resveratrol, and cysteamine, on the maturation and developmental capacity of yak oocytes in vitro. The results showed that adding melatonin to the oocyte maturation solution significantly increased the maturation rate and the proportion of mitochondrial homogenous distribution of yak oocytes. The oocyte cleavage rate of in-vitro-fertilized embryos also increased. Among these three antioxidants, cysteamine had the strongest effect on the increase of GSH content in oocytes [100].

### 4.3. Microelement

Zinc (Zn) is an essential trace element in mammals that plays an important role in cell growth, proliferation, division, and immunity [101,102]. Xiong Xianrong et al. discovered that adding 2 mg/L zinc sulfate to the IVM medium of yak oocytes increased glutathione (GSH) content, superoxide dismutase (SOD) activity, and the blastocyst rate, and significantly reduced ROS levels. This effect could be achieved by Zn^2+^ inducing the up-regulation of *Zn transporters 3* (*ZnT3)*, *Zrt*, and *Irt-like protein 14* (*ZiP14*) expressions in yak oocytes [103]. Hu Jiajia et al. obtained similar results in their study. Moreover, they discovered that adding 2 mg/L zinc sulfate significantly increased the expression levels of *Solute-linked carrier* (*SLC30A*) and *SLC39A* family members, including *SCL30A3*, *SLC30A6*, *SCL30A9*, *SLC39A6* and *SLC39A14* in yak mature oocytes, facilitating the cleavage of the fertilized ovum and blastocyst formation [104]. Feng Yun et al. revealed that adding 0.8 mg/L zinc sulfate could improve the yak oocyte maturation and the efficiency of in vitro fertilization by increasing the antioxidant enzyme gene (*SOD1*, *CAT*, *TXN1*, and *PRD1*) expression levels, and also up-regulating the cumulus cell expansion related genes (*PTX3* and *TSG6*) in the oocyte [105]. 

Calcium (Ca) is an important second messenger in cells, and the changes in Ca^2+^ concentration are closely related to regulating physiological functions by affecting signal transduction [106]. Chen demonstrated that adding Ca^2+^ at 0.24 mM in yak oocyte culture medium significantly increased the oocyte maturation rate in vitro. The mechanism was that Ca^2+^ activated the activity of calmodulin-dependent protein kinaseII (CaMKII), increased GSH content, and decreased the ROS level in the oocyte. Ca^2+^ up-regulated the expressions of *BCL-2*, *EGF*, *EGFR*, and *C-FOS*, whereas it down-regulated *BAX* expression [107].

Selenium (Se) is an essential trace element for reproduction, immunity, antioxidant systems, embryonic growth, and other physiological functions [108,109,110]. Xiong Xianrong et al. observed that 2 μg/mL of sodium selenite significantly increased the glutathione peroxidase (GSH-Px) activity in the oocytes and the blastocyst rate of subsequent embryos by adding sodium selenite to the in vitro culture medium of yak COCs. Those effects were achieved by increasing the selenoprotein synthesis-related gene expression levels, including *GPX4*, *SEPP1*, *RPL22*, and *CCND1* in oocytes and cumulus cells [111].

### 4.4. Small Molecule Compounds

Cyclic adenosine monophosphate (cAMP) is the first discovered second messenger. cAMP can regulate the various target genes’ transcription, primarily via protein kinase A (PKA) and its downstream effectors [112]. cAMP plays a key role in maintaining oocyte meiotic arrest and initiating meiotic resumption in mammalian oocytes [113,114]. Previous studies have demonstrated that maintaining cAMP levels in oocytes before oocyte maturation could temporarily repress spontaneous meiotic resumption, thereby improving oocyte developmental competence and subsequent embryonic development [115,116,117,118]. Xiong Xianrong et al. revealed that a supplement with a cAMP activator, cilostazol, would benefit yak oocytes IVM by increasing cAMP and GSH levels and modulating mRNA expression patterns during pre-IVM. Specifically, adding cilostazol to the in vitro maturation medium and pre-IVM for 2 h or 4 h significantly increased the *PKA1* and *CY3* mRNA expression levels. It also significantly decreased the *PDE3A* mRNA expression level in yak COCs and blastocysts [119]. 

Roscovitine and C-type natriuretic peptide (CNP) are two meiotic arrest factors promoting yak oocyte maturation in vitro. Roscovitine is a member of the 2,6,9-trisubstituted purine family and has a structure similar to ATP. Therefore, roscovitine interacts with amino acids in the ATP-binding pocket of the catalytic domain of some Cyclin-dependent kinases (CDK), preventing ATP from binding to CDK, inhibiting CDK activity, and ultimately blocking the cell cycle [120]. Pretreatment of yak COCs with 12.5 μM roscovitine for 6 h followed by conventional IVM improved the quality of yak mature oocytes. Liu Yu counted the cell expansion index (CEI) of ovarian thalamus granulosa cells after treating pre-IVM yak COCs with different concentrations of roscovitine. They discovered that pretreatment with 12.5 μM roscovitine for 6 h significantly increased the CEI of COCs. This treatment significantly reduced the ROS content in oocytes, promoted the uniform distribution of mitochondria, and enhanced the structure of transzonal projections (TZPs), thereby improving the quality of yak oocytes. Furthermore, this treatment significantly up-regulated the mRNA expressions of antioxidant gene *SOD2*, anti-apoptotic gene *BCL-2*, and development-related genes *GDF9*, *EGFR*, and *ZAR1*, and significantly down-regulated the mRNA expression of the pro-apoptotic gene *BAX* [121]. 

CNP is a natural determinant of meiotic arrest that can maintain gap junction activity and support the key gene expressions essential for oocyte development [122]. Jing Tian counted the number of yak GV oocytes cultured with different concentrations of CNP and discovered that the quality, maturation rate, and blastocyst rate of yak oocytes could be significantly improved by pre-IVM of oocytes at 100 nM CNP for 6 h and IVM for 28 h. This treatment significantly increased TZP and GSH protein expressions and decreased ROS levels in yak oocytes. These effects might be due to CNP significantly promoting the expressions of CNP receptor gene *NPR2*, anti-apoptotic gene *BCL-2*, and growth differentiation factor *GDF9* in oocytes and blastocysts. CNP suppressed the *EGF* and its receptor *EGFR* gene expressions in oocytes while promoting *EGF*, *EGFR*, and *DNMT1* expressions in blastocysts and repressing the expression of pro-apoptotic gene *BAX* in blastocysts [123].

Current studies on mouse, pig, and buffalo oocytes indicate that glucose is an important energy source for oocyte maturation [124,125,126,127]. Sucrose plays an important role in oocyte development, restoration of oocyte aging, and oocyte cryopreservation [128,129,130]. Shi Xian et al. discovered that an in vitro maturation medium containing 10 mmol/L glucose could significantly increase the nucleus maturation rate of yak oocytes and the cleavage rate of in vitro fertilization embryos. They also indicated that a medium containing 10 mmol/L sucrose could significantly increase the nucleus maturation rate of yak oocytes [131].

### 4.5. Hormones

Xiao Xiao et al. demonstrated that supplementing IVM medium with 5 μg/mL FSH and 50 IU/mL LH improved the developmental ability of yak oocytes after in vitro fertilization (IVF) [132]. He Honghong et al. studied the effects of different concentrations of FSH concentrations on yak oocytes and revealed that the oocyte maturation rate was highest in the 5 μg/mL FSH treatment group. Further study indicated that FSH might improve yak oocyte development by increasing *EGF* and *EGFR* mRNA expression levels, and it might inhibit oocyte apoptosis by increasing antiapoptotic gene *BCL-2* expression while reducing pro-apoptotic gene *BAX* expression [133]. 

Estradiol (E2) is the most active and predominant maternal estrogen during pregnancy [134]. A supplement of exogenous E2 or promoting endogenous E2 synthesis and secretion can improve oocyte maturation and increase cumulus cell spread during oocyte maturation in various animals [135,136,137,138,139]. Pan Yangyang et al. discovered that adding 10^−4^ mM endogenous 17β-estradiol to the IVM medium of COCs could increase the cumulus expansion and subsequent oocyte development. These might result from increasing the expressions of cumulus-expansion-related factors (*HAS2*, *PTGS2*, and *PTX3*) and the oocyte-secreted factors (*GDF9*, *FGF10*, and *BMP15*) [140]. 

Rfamide-related peptides 3 (RFRP-3), a structural and functional homolog of gonadotropin-inhibiting hormone (GnIH), has been proposed as a new breeding inhibitory neurohormonal peptide that plays a crucial role in the reproductive axis across various species [141,142]. Xiong Xianrong et al. showed that RFRP-3 might inhibit yak oocyte maturation and developmental potential by binding with its receptor GPR147. Specifically, high-dose (10^−6^ mol/L) RFRP-3 inhibited yak oocyte proliferation, induced oocyte apoptosis, and decreased 17β-estradiol and progesterone (P4) concentrations. RFRP-3 dose-dependently elevated the expression of apoptosis markers (*Caspase* and *Bax*), whereas the expression levels of steroidogenesis-related factors (*LHR*, *StAR*, *3b-HSD*) were down-regulated in a dose-dependent manner [143].

Cytochrome P450arom (CYP19A1) is the key enzyme for gonadal hormone synthesis in most animals [144]. CYP19A1 up-regulated the endogenous E2 level and enhanced the developmental ability of yak oocytes. Specifically, the treatment of CYP19A1 activator AFB1 up-regulated the endogenous E2 level and increased the rates of IVM and blastocysts, while decreasing the E2 level. IVM and blastocyst rates were observed in the CYP19A1 inhibitor BPA treatment group, which implied that CYP19A1 played an essential role in oocyte and embryo development of yaks [145].

### 4.6. Platelet-Activating Factor (PAF)

PAF is an acetylated glycerol signaling phospholipid important physiological regulator in reproduction [146]. PAF exerts its actions via activating specific PAF receptors (PAF-R) in cells and tissues of the female reproductive tract [147]. Wang Qin explored the role of PAF in the maturation of yak oocytes and early embryonic development by adding different concentrations of PAF to the maturation medium of COCs in vitro. The results showed that 10^−7^ mol/L PFA significantly increased the maturation, cleavage, and blastocyst rates of yak oocytes by regulating *BAX*, *BCL-2*, *EGF*, *EGFR*, *C-FOS*, *OCT-4*, and *NANOG* gene expressions [148,149].

The effects of some exogenous factors on yak oocytes and the possible mechanisms are shown in Table 2.

## 5. Environment Factor

### 5.1. Temperature

Pang Bo et al. investigated the effects of ovarian preservation temperature and the culture methods on the maturation rate of yak oocytes in vitro, and the results showed that preserving the ovaries from 20 °C to 25 °C could improve the oocyte maturation rate in vitro [150]. Ma Li et al. preserved fresh yak ovaries in saline at different temperatures (15–20 °C; 25–30 °C; 35–40 °C). The results indicated that 25–30 °C was the optimal temperature for the ovarian transport of yaks. The maturation rate of oocytes, eight-cell embryo formation, and blastocyst rates of IVM and parthenogenetic activation (PA) embryos were significantly higher than those of other groups within this temperature range [151].

### 5.2. Oxygen

Oxygen (O_2_) is vital to maintain and complete oocyte maturation and embryonic development. Changes in oxygen concentration during oocyte maturation in vitro affect nuclear DNA methylation, intracellular reactive oxygen species (ROS) levels, and cellular aging [87,152]. Low oxygen levels are the naturally preferred microenvironment for most processes during early development and mainly drive proliferation [153,154]. Several studies proved that culturing oocytes and embryos under low oxygen conditions improved their developmental capacity [155,156,157].

He Honghong et al. investigated the development of yak oocytes and preimplantation embryos under low oxygen concentrations. The results showed that 5% oxygen facilitated yak oocyte maturation in vitro and significantly increased the cleavage and blastocyst rates of the subsequent PA embryos. The blastocyst rate of IVM embryos was significantly increased at the same oxygen concentration. The proper mechanism might be that 5% oxygen up-regulated *HIF-1α* expression. *HIF-1α* subsequently increased the *BCL-2* mRNA expression and decreased the *BAX* mRNA expression in yak COCs and blastocysts, whereas *CDX2*, *POU5F1*, *SOX2*, and *NANOG* mRNA expressions in blastocysts were significantly increased [158,159,160]. 

Li Ruizhe compared the expression differences in the transcriptome of yak oocytes at 5% and 20% oxygen concentrations and revealed that the genes up-regulated in the 5% group were mainly involved in hypoxia response, the cell cycle, chromatin conformation and remodeling, and the cytoskeleton, including the *WDR26*, *MKP2K1*, *MAPK1*, *TICRR*, *WAC*, *EIF4ENIF1*, *ODC1*, *CHAMP1*, *MKI67*, *MCM10*, *SFMBT1*, *PBRM1*, *KAT8*, *IQGAP2*, *EPS8*, and *RANBP9* genes. The genes up-regulated in the 20% oxygen concentration group, including *ACAT1*, *ATP5MF*, *AURKAIP1*, *COX6A1*, *NDUFA10*, *NDUFA11*, *NDUFS7*, *LYRM7*, *UQCR10*, *EIF1AX*, *RPL13*, *RPL13A*, *RPL34*, *RPLP2*, *GSTA2*, *QARS*, *NOSTRIN*, *SH3BGRL3*, *TIMP1*, *S100A11*, and *PTX3*, were primarily involved in energy metabolism, protein synthesis, redox homeostasis, and oocyte regulation [100]. Additionally, Li Ruizhe et al. discovered that 5% O_2_ increased the oocyte maturation rate and GSH content, decreased the oocyte ROS level, and improved the quality of PA and IVF blastocysts. The effects were achieved by decreasing the expressions of the antioxidant genes *CAT* and *GPX1*, increasing the expression of the metabolism-related gene *LDHA*, and embryo development-related genes *CDX2* and *OCT4* in yak IVF blastocysts [100,161].

## 6. Discussion

Various endogenous, exogenous and environmental factors that regulate yak oocyte and embryo development in vitro have also been reported to play important roles in regulating bovine oocyte and embryo development [162,163,164,165,166,167,168,169,170,171,172,173,174,175,176,177,178,179,180,181].

However, the concentration or molecular mechanism of these factors to exert their optimal effects on oocyte and embryo development in bovines are somewhat different from those in yaks. For example, the optimal concentrations of FGF10 and LIF in promoting oocyte and embryo development in yaks were 5 ng/mL and 50 ng/mL [6,84], respectively, while in bovines, the optimum concentrations of these two factors were 2.5 ng/mL and 25 ng/mL, respectively [164,165]. In yaks, melatonin reduces the level of intracellular ROS through its antioxidant activity, thus reducing cellular ROS levels, inhibiting cell apoptosis and increasing the oocyte maturation rate and IVF embryo cleavage and blastocyst rates [98,99], however, in bovines, melatonin mainly promotes oocyte and embryo development in vitro by up-regulating the expression of GJA4 to enhance gap junction intercellular communication [166]. Whether these differences are due to differences between species needs further study.

The number of factors that regulate oocyte and embryo development in yaks is still far less than that in traditional domestic animals, such as bovines and pigs. Furthermore, current studies mainly focus on the effects of different factors in the maturation and developmental competence of yak oocytes in vitro; little is known about the related molecular mechanisms. As the yak is an atypical animal living on the plateau, the difficulty of sample collection brought great obstacles to the preparation of experimental materials. In addition, the individual differences between different yaks greatly interfere with the accuracy of experimental results. These problems need to be solved urgently in the field of yak breeding in the future.

## 7. Conclusions

High-quality oocytes are the basis of the application of assisted reproductive technology in livestock breeding. The various factors that influence the maturation and developmental competence of yak oocytes in vitro summarized here will help researchers to further optimize the conditions of IVM of yak oocytes in vitro, thus promoting the improvement of yak breeding and the development of animal husbandry in the plateau area. 

## Figures and Tables

**Table 1 genes-14-01882-t001:** Effects of some of exogenous factors on in vitro maturation and development of yak oocytes.

Endogenous Factor		Model	Treatment	Effect	Possible Mechanism	Reference
GPR50	-	Yak oocyte	Knockdown GPR50	↓ Maturation and polar body 1 excretion rates	-	[62,63]
Histone modifications	KDM1A	Yak COCs	Adding GSK-KDM1A, the specific inhibitor of KDM1A	↓ Cumulus cell expansion, the first polar body excretion rate, and cleavage	↑ OCT4 ↑ SOX2	[47,48]
	KDM2	Yak oocyte	Adding butyrazide, a specific inhibitor of KDM2	↓ Oocyte maturation ↑ Cleavage and blastocyst rates	-	[50]
	SIRT1	Yak COCs	Adding the specific agonist SRT2104	↑ Cumulus cell expansion, cleavage and blastocyst rates ↑ Early embryos’ development ability ↓ Oocyte aging ↓ ROS level	↑ SIRT1 ↑ FOXO3a ↑ SOD2 ↓ BAX	[53]
			Adding the inhibitor inauhzin	↓ Cumulus cell expansion, cleavage and blastocyst rates	-	
miRNA	miR-342-3p	Yak oocyte	MiR-342-3p inhibitor	↓ Oocyte meiotic maturation	MiR-342-3p inhibitor up-regulated DNMT1 expression	[60]

↑: increases. ↓: decreases.

**Table 2 genes-14-01882-t002:** Effects of endogenous factors on in vitro maturation and development of yak oocytes.

Exogenous Factor		Sample	Treatment	Effect	Possible Mechanism	Reference
Growth factor	EGF	Yak COCs	Adding 100 ng/mL EGF	↑ Maturation rate ↑ Cleavage and blastocyst rates	↓ BAX ↑ BI-1	[73]
		Yak COCs	Adding 40 μg/mL EGF	↑ Maturation rate ↑ PA embryos’ development ability	-	[72]
	IGF-1	Yak oocyte	Adding 100 ng/mL IGF-1	↑ Maturation rate ↑ Cleavage and blastocyst rates	↑ CIRP	[76]
	FGF10	Yak COCs	Adding 5 ng/mL FGF10	↑ Maturation rate ↑ Fertilization ability	↑ CD9 ↑ CD81 ↑ DNMT1 ↑ DNMT3B	[6]
	LIF	Yak oocyte	Adding 50 ng/mL LIF	↑ Maturation rate ↓ ROS generation ↓ Apoptosis levels ↑ Blastocyst total cell numbers ↓ Blastocyst apoptosis rate	↑ BCL2 ↑ CAPASE3 ↑ SURVIVIN ↑ SOD2 ↑ GPX4	[84]
Antioxidants	Melatonin	Yak COCs	Adding 10^−9^ M melatonin	↑ Maturation rate ↑ Cleavage and blastocyst rates ↑ GSH content ↓ ROS levels ↓ Mitochondrial proteins and DNA damage extent ↓ oocyte and blastocyst apoptosis	-	[98,99,100]
	Vitamin C	Yak COCs	Adding 50 μg/mL AA to IVM medium exposed to AFB1	↓ ROS level ↓ Early oocyte apoptosis ↑ Mitochondrial distribution status ↑ DNA methylation Restored actin distribution	↑ DNMT3a ↑ TET3 ↑ NANOG ↑ POU5F1 ↑ CDX2 ↓ DNMT1	[38,100]
	Vitamin A	Yak oocyte	Adding 2 μM vitamin A	↑ Maturation rate ↑ PA embryo cleavage rate	↑ STRA8 ↑ RARA ↑ RXRA	[93,94]
	Resveratrol	Yak COCs	-	↓ ROS levels ↑ GSH content	-	[100]
	Cysteamine	Yak COCs	-	↑ GSH content	-	[100]
Microelement	Zinc	Yak COCs	Adding 2 mg/L zinc sulfate	↑ GSH content ↑ SOD activity ↑ Cleavage and blastocyst rates ↓ ROS levels	↑ ZnT3 ↑ Zrt ↑ ZiP14 ↑ SLC30A3 ↑ SLC30A6 ↑ SCL30A9 ↑ SLC39A6 ↑ SLC39A14	[103,104]
			Adding 0.8 mg/L Zinc sulfate	-	↑ SOD1 ↑ CAT ↑ TXN1 ↑ PRD1 ↑ PTX3 ↑ TSG6	[105]
	Calcium	Yak oocyte	Adding 0.24 mM Ca^2+^	↑ Maturation rate ↑ GSH content ↓ ROS levels	↑ activity of CaMKⅡ ↑ BCL-2 ↑ EGF ↑ EGFR ↑ C-FOS ↓ BAX	[107]
	Selenium	Yak COCs	Adding 2 μg/mL sodium selenite	↑ GSH-Px activity ↑ Blastocysts rate	↑ GPX4 ↑ SEPP1 ↑ RPL22 ↑ CCND1	[111]
Small molecule compounds	Roscovitine	Yak COCs	Pre-IVM yak COCs with 12.5 μM roscovitine for 6 h	↑ CEI of COCs ↑ Uniform distribution of mitochondria ↑ TZPs ↓ ROS levels	↑ SOD2 ↑ BCL- ↑ GDF9 ↑ EGFR ↑ ZAR1 e ↓ BAX	[121]
	CNP	Yak oocyte	Pre-IVM yak oocytes with 100 nM CNP for 6 h and IVM for 28 h	↑ TZPs ↑ GSH content ↓ ROS levels	↑ NPR2 ↑ BCL-2 ↑ GDF9 ↑ EGF ↑ EGFR ↑ DNMT1 ↓ BAX	[123]
	Glucose	Yak oocyte	Adding 10 mmol/L glucose	↑ Maturation rate ↑ Cleavage rate	-	[131]
	Sucrose	Yak oocyte	Adding 10 mmol/L sucrose	↑ Nucleus maturation rate	-	[131]
Hormone	FSH	Yak oocyte	Adding 5 μg/mL FSH	↑ Maturation rate ↑ IVF embryos’ development ability ↓ Oocyte apoptosis	↑ EFG ↑ EGFR ↑ BCL-2 ↓ BAX	[132,133]
	LH	Yak oocyte	Adding 50 IU/mL LH	↑ IVF embryos’ development ability	-	[132]
	RFRP-3	Yak COCs	Adding 10^−6^ mol/L RFRP-3	↑ Oocyte proliferation ↑ Oocyte apoptosis ↓ E2 and P4 concentrations	↑ CASPASE ↑ BAX ↓ LHR ↓ StAR ↓ 3b-HSD	[143]
	E2	Yak COCs	Adding 10^−4^ mM endogenous 17β-estradiol	↑ Cumulus expansion ↑ Cleavage and blastocyst rates	↑ HAS2 ↑ PTGS2 ↑ PTX3 ↑ GDF9 ↑ FGF10 ↑ BMP15	[140]
	CYP19A1	Yak COCs	Adding CYP19A1 inducer (AFB1)	↑ E2 level ↑ Autophagy level ↑ Maturation rate ↑ Cleavage rate	↑ CYP19A1 ↑ ATG5 ↑ BECLIN1 ↑ LC3	[144,145]
			Adding CYP19A1 inhibitor (BPA)	↓ E2 level ↓ Autophagy level	↓ CYP19A1 ↓ ATG5 ↓ BECLIN1 ↓ LC3	
PAF	-	Yak COCs	Adding 10^−7^ mol/L PFA	↑ Maturation rate ↑ Cleavage and blastocyst rates ↓ Apoptosis level	↑ BCL-2 ↑ EGF ↑ EGFR ↑ C-FOS ↑ OCT-4 ↑ NANOG ↓ BAX	[148,149]

↑: increases. ↓: decreases.

## Data Availability

No new data were created or analyzed in this study. Data sharing is not applicable to this article.

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
