# Peer review of "Factors Influencing the Maturation and Developmental Competence of Yak (Bos grunniens) Oocytes In Vitro"

_genes, 2023, doi:10.3390/genes14101882_

Round 1

Reviewer 1 Report

An expensive coverage has been given by authors on factors of any kind that could determine the quality of Yak oocytes that may be subjected to in vitro fertilization. Some parts of your paper are redundant, see for example paragraph 2.2, in which the information given by authors on COCs layers of granulosa cells, has alreaby been widely described. A simple reference could do it.

Written English should be revised with professional help

Reviewer 2 Report

The review work by Mo et al., describes the maturation and developmental competence of yak (Bos grunniens) oocytes in vitro. The review is detailed and timely. The authors described almost every factor studied in Yak oocyte maturation, which is commendable.

However, one major comment that I would like to do is that in some instances reading the review sounded like a laundry list of experiments that are being described there without any comments or implications or perspective from the authors. Meaning that it would be great if the authors put their point of view after each section stating where the field is moving. Similarly putting some key questions will be more attractive and engaging.

Minor:

L41-45: Requires reference. Use the review (Das and Arur, 2022, Regulation of oocyte maturation: Role of conserved ERK signaling) as reference.

Section 2.2. A pictorial illustration will make the concept.

Reviewer 3 Report

General and specific comments

Dear Authors,

1.       Please go through your manuscript acknowledging a more balanced view. I would suggest more balanced wording. Many good ideas, but often presented without the required balance. Please, make your statements more neutral. Some paragraphs are blurring the message.

2.       A second major problem is that you are mixing very often different ideas. The document should be rendered more rigorous and organized. Also, it would make sense to give a very rigorous and theoretical sound explanation of what factors influence the maturation and developmental competence in yaks including oocytes and embryos.

3.       The concept of maturation and developmental competence of oocytes and embryos might need a more theoretical description. All is correct BUT needs organization. Suggestions: - Introduce one topic after the other - Keep a logical path of thoughts - Group ideas of authors to clusters in a review this is the added value.

4.       Regarding oocyte and embryo metabolism (which appears in several places throughout the manuscript, e.g. Line 109, Line 122, Line 279, Lines 512-513) please put adequate references found in the literature.

5.       Line 151: Interesting and correct thought. This may be the reason why you insist that information beyond the DNA sequence can be inherited from parents to offspring then the “improved” genetic evaluations broaden the genetic base. My suggestions: - Good idea, please keep it - But put it elsewhere, the current place is suboptimal.

6.       I suggest adding a new section (before the conclusion section) to complete the present review comparing yaks with other bovine species with regard to the factors influencing the maturation and developmental competence of oocytes and embryos.

In conclusion, please revise your manuscript following the suggestions. Currently, the lack of clear organization and some important references is reducing its usefulness. 

Moderate editing of English language required.

(please add the following references as well:  https://doi.org/10.1071/RD11080  and https://doi.org/10.1071/RD17429 )

- In general you should have a paragraph where you link in using the following reference https://doi.org/10.3390/ani11030599

I suggest you include the following reference in this new section: a) https://doi.org/10.1095/biolreprod.110.087411 . It has elements and references that seem to be missing in your review

Round 2

Reviewer 2 Report

The provided revised version is sufficiently improved to warrant publication in Genes.

Reviewer 3 Report

Dear authors,

Thank you very much for the corrections provided. The manuscript can now be accepted after minor editing of English language.

Minor editing of English  language required.